# Determinants of Depressive Symptoms in People Living with HIV: Findings from a Population-Based Study with a Gender Perspective

**DOI:** 10.3390/ijerph20043687

**Published:** 2023-02-19

**Authors:** Ivet Bayes-Marin, Laia Egea-Cortés, Jorge Palacio-Vieira, Andreu Bruguera, Jocelyn Mesías-Gazmuri, Josep M. Llibre, Emma Fernández, Arkaitz Imaz, Carlos G. Forero, Cristina Agustí, Laia Arbones-Fernández, José M. Miró, Jordi Casabona, Juliana Reyes-Ureña

**Affiliations:** 1Department of Medicine, School of Medicine and Health Sciences, Universitat Internacional de Catalunya, 08195 Sant Cugat del Vallés, Spain; 2Centro de Investigación Biomédica en Red de Salud Mental (CIBERSAM), Instituto de Salud Carlos III, 28029 Madrid, Spain; 3Centre of Epidemiological Studies of HIV/AIDS and STI of Catalonia (CEEISCAT), Health Department, Generalitat de Catalunya, 08916 Barcelona, Spain; 4Germans Trias i Pujol Research Institute (IGTP), Campus Can Ruti, 08916 Badalona, Spain; 5Centro de Investigación Biomédica en Epidemiologia y Salud Pública (CIBERESP), 28029 Madrid, Spain; 6Department of Pediatrics, Obstetrics and Gynecology and Preventive Medicine, Universidad Autónoma de Barcelona, 08193 Barcelona, Spain; 7Fight AIDS and Infectious Diseases Foundation, 08916 Badalona, Spain; 8Infectious Diseases Service, Hospital Clínic-IDIBAPS, University of Barcelona, 08036 Barcelona, Spain; 9Hospital Universitari de Bellvitge, Bellvitge Biomedical Research Institute (IDIBELL), University of Barcelona, L’Hospitalet de Llobregat, 08908 Barcelona, Spain; 10Internal Medicine Department, Hospital de Mataró, 08304 Barcelona, Spain; 11Centro de Investigación Biomédica en Enfermedades Infecciosas (CIBERINFEC), Instituto de Salud Carlos III, 28029 Madrid, Spain

**Keywords:** depressive symptoms, health-related quality of life, HIV/AIDS, gender, transgender

## Abstract

Depressive symptoms are common among people living with HIV (PLWH). The aim of this study was to identify the determinants of depressive symptoms in PLWH in Spain. A total of 1060 PLWH participated in this cross-sectional study and completed the Patient Health Questionnaire-9. The odds ratios for the presence of depressive symptoms were analyzed in a multivariable logistic regression model, including sociodemographic data, comorbidities, health-related behaviors, and social-environment-related variables. We found an overall prevalence of depressive symptoms of 21.42%; by subgroup, namely men, women, and transgender persons, prevalence was 18.13%, 32.81%, and 37.14%, respectively. Moreover, social isolation (OR = 1.05 [CI, 1.02–1.08]) and poor physical and mental quality of life (OR = 1.06 [CI, 1.02–1.09] and OR = 1.13 [CI, 1.09–1.17], respectively) were associated with depressive symptoms. As protective factors, we identified serodisclosure to more people (vs. none; OR = 0.39 [CI, 0.17–0.87]), satisfaction with social roles (OR = 0.86 [CI, 0.79–0.94]), better cognitive function (OR = 0.92 [CI, 0.89–0.95]), and sexualized drug use once in a lifetime (OR = 0.52 [CI, 0.29–0.93]). This study showed a high prevalence of depressive symptoms in PLWH, especially among women and transgender people. The association between psychosocial variables and depressive symptoms highlights the multidimensionality of the problem and identifies areas for intervention. This study found that the management of mental health issues is an area that needs to be improved and tailored to specific groups, with the aim of enhancing the well-being of PLWH.

## 1. Introduction

In 2021, approximately 160,000 people were estimated to be living with HIV in Spain; the estimated prevalence of HIV in persons aged 15 to 49 years (0.3%) was similar to that reported in other European countries with concentrated epidemics, such as France (0.3%) and Italy (0.3%) [1]. At the end of 2019, it was estimated that 87% of people living with HIV (PLWH) were diagnosed, 97.3% of those diagnosed were receiving ART, and viral load was suppressed in 90.4% of those receiving ART [2]. Access to the Spanish healthcare system is universal and free of charge for all citizens. Individuals not born in Spain are also entitled to healthcare under the same conditions as Spanish citizens. In Spain, treatment for HIV is provided exclusively through hospital pharmacies; therefore, most HIV-infected patients receive HIV care and treatment in public hospitals [3]. HIV patients attend the HIV clinic for routine checkups every 6 months if the infection is under control. Mental healthcare is included as part of routine follow-up.

The success of antiretroviral therapy (ART) has lessened the clinical complications of HIV infection and increased the survival rate of PLWH, with the infection now considered a chronic disease in those countries where treatment is accessible [4,5]. Nevertheless, previous research indicated that PLWH have a worse health-related quality of life (HRQoL) than the general population [6,7,8]. Most of these studies considered HRQoL to be a multidimensional construct comprising factors such as physical, emotional, cognitive, and social functioning [6,7]. Using validated instruments, they found worse HRQoL in PLWH in all the domains, particularly anxiety and depression, but also poor self-rated health status, than in people without HIV [6,7]. Insight into HRQoL is crucial if we are to understand the effects of HIV as a chronic disease and address the unmet needs of this population. However, few studies on HRQoL in PLWH have been carried out in the Spanish population, and samples are often non-representative [5,7,8,9]. 

As stated above, mood disorders are clearly related to poorer HRQoL in PLWH. Depression is the most common mental health problem, being two to three times more common in PLWH than in the general population [10,11,12,13,14]. Depression is considered a leading cause of disability worldwide and a major contributor to disease burden worldwide [15]. In the case of HIV, depression becomes an even more relevant target for intervention, since it emerges as a public health issue owing to its multiple consequences for well-being, HRQoL, HIV management, and prevention of transmission. 

The multiple factors associated with the high prevalence of depression in this population point to a complex and multicausal relationship. These include demographic and economic factors (e.g., female sex) [16,17], low level of education [17,18], and financial instability [19]. Additional, overlapping risk factors include pre-existing mood disorders and harmful alcohol consumption [20,21], substance abuse [22], nicotine dependence [19], poor adherence to ART [18], risky sexual behaviors [22], structural barriers such as internalized stigma [18], experiences of discrimination [18,23], and social isolation [21,23].

Women living with HIV (WLWH) are more likely to experience mental health disorders and other comorbidities than men and non-HIV-infected women [24]. The fact that WLWH experience systematic structural barriers might lead to mental health impairment, which can be aggravated by HIV-related disparities associated with gender, race/ethnicity, poverty, and rural location, along with intersectionality [24]. Moreover, transgender people are prone to poorer self-reported mental health than cisgender women and men living with HIV [25]. However, these factors have received little attention in the literature, as most studies and interventions have focused mainly on men who have sex with men (MSM) [26]. Therefore, there is a need to investigate gender group differences by considering the multiple factors associated with mental health and gender so that the problems affecting this population can be addressed in a timely manner [26].

The objective of the present study was to identify risk and protective factors for sociodemographic variables, comorbidities, health-related behaviors, and social-environment-related variables associated with depressive symptoms in PLWH, taking into account gender differences, in order to enable clinical and public health interventions.

## 2. Materials and Methods

### 2.1. Study Design and Sample Recruitment

Vive+ is a cross-sectional study that ran from October 2019 to March 2020 and was nested in the PISCIS cohort. The PISCIS cohort is described elsewhere [27,28]. Briefly, it is an HIV cohort that has been prospectively followed up since 1998. It comprises PLWH who are receiving care in 16 hospitals in Catalonia and 2 in the Balearic Islands (Spain). Based on the population in follow-up in the cohort in 2017 (n = 14,190 patients), and taking into account an expected prevalence of depression and anxiety of 30% [29,30] and a type 1 error of 5% and study power of 95%, we calculated a sample size of 1186 PLWH.

Patients were invited to participate at the HIV units by a trained peer during a regular visit. Patients were excluded if they did not speak Spanish, were unable to complete the questionnaire by themselves owing to mental disability, or did not agree to sign the consent form. According to the number of PLWH in follow-up in each HIV unit at the time of the study, a sample size was assigned at each hospital. We over-sampled people aged >60 years and women in order to obtain a sufficiently large sample with these demographic characteristics to adjust the results. All participants who agreed to participate signed a consent form.

The questionnaire was a piloted, structured, ad hoc, and self-administered instrument that was completed using a tablet. Data from the questionnaire were anonymized and linked back with the PISCIS cohort identification number to retrieve information related to HIV-related clinical care.

This study was performed in line with the principles of the Declaration of Helsinki. Ethics approval for the study was obtained from the Ethics Committee of the Germans Trias i Pujol Hospital (No. PI-19-172), and permission to conduct the study in the all the health facilities was granted by each local ethics committee.

### 2.2. Variables

#### 2.2.1. Outcome

To assess depressive symptoms, we used the Patient Health Questionnaire-9 (PHQ-9) [31,32], which comprises 9 items based on the Diagnostic and Statistical Manual of Mental Disorders (DSM-IV) criteria. PHQ-9 is a reliable tool that has been validated for PLWH [33,34]. Participants were asked to indicate how often they have been bothered by 9 possible symptoms in the last 2 weeks (i.e., “feeling down, depressed, or hopeless”; “feeling tired or having little energy”). The results were scored as follows: 0, “not at all”; 1, “several days”; 2, “more than half the days”; and 3, “nearly every day”. Scores were summed to obtain scale scores, ranging from 0 to 27. Scores of 5, 10, 15, and 20 represent the cutoff points for mild, moderate, moderately severe, and severe depressive symptoms, respectively [31]. In our analyses, we coded depressive symptoms as no (none or mild) or yes (moderate, moderately severe, or severe depressive symptoms). Cronbach’s α coefficient was 0.91.

#### 2.2.2. Predictors

##### Sociodemographic Variables

In this block, self-reported gender was coded as men, women, or transgender. We combined transgender men and transgender women, owing to the small number of people in the sample. Age was categorized as ≤39, from 40 to 59, and ≥60 years. We also considered whether participants were born abroad (yes/no), their level of education (none or primary school, secondary school, or higher education), occupation (currently working, unemployed, retired, homemaker, or on sick leave), and monthly income (no income, <EUR 1000, EUR 1001–2000, or >EUR 2001). Sexual attraction was coded as heterosexual, homosexual (gay, lesbian), bisexual, or other. Finally, recent sex work (i.e., have been paid for sex in the previous 6 months) was considered a binary indicator (yes/no).

##### Comorbidities

Several variables related to comorbid status were included. HRQoL was evaluated using the 12-Item Short Form Survey (SF-12), a freely distributed questionnaire [35]. The SF-12 assesses functional status in 2 domains, namely physical and mental health [36], and comprises 12 items scored using 3- or 5-point Likert-type response options. To calculate the scores, we applied the Bidimensional Response Process Model Algorithm (BRP-12) [37], which is based on the item-response theory [38], and obtained a set of scores that directly mapped into the SF-12. In the case of the BRP, higher scores are indicative of poor health.

Cognitive function was assessed using the Neuro-QOL Item Bank v2.0—Cognitive Function [39]. The Neuro-QOL comprises a pool of validated instruments based on self-reported measures to evaluate HRQoL in adults and children. The time frame is “the last 7 days”, and each of the 8 items that constitute the abbreviated versions have 5 response options: 1, “never”; 2, “rarely”; 3, “sometimes”; 4, “often”; and 5, “always”. The total score is obtained by adding the scores of the individual items, resulting in a minimum score of 8 and a maximum score of 40. In the cognitive function measure, higher scores mean better cognitive functioning.

We also asked the participants if they were satisfied (“very satisfied”, “satisfied”, “unsatisfied”, or “very unsatisfied”) with their sex life. These labels were coded into a dichotomized variable (satisfied/unsatisfied).

Alcohol dependence was assessed using the Alcohol Use Disorders Identification Test (AUDIT-C) [40], a self-administered screening questionnaire comprising 3 items that are used to estimate alcohol abuse or dependence. Scores range from 0 to 12, with 0 indicating no alcohol consumption. The cutoff points differ between men and women (4 or more points for women and 5 or more for men) [41]. Since there are no specific cutoff points for transgender people, we used biological sex to determine whether a participant was a risky drinker. Finally, 3 categories were defined based on the abovementioned cutoff points: non-drinker, low-risk drinker, and high-risk drinker.

Information regarding the amount of time since HIV diagnosis was included in the analysis, and for the present study, we considered whether the participant had been diagnosed with HIV infection in the previous 12 months (yes/no). Furthermore, the mode of transmission was considered and included in our analyses as follows: through injecting drugs (people who inject drugs, PWID) and through sexual intercourse between MSM or heterosexual relations (between men who have sex with women, HHTX; or women who have sex with men, MHTX).

##### Health-Related Behaviors

In terms of substance use, participants were asked about their consumption of illegal drugs in the previous year and previous month or their daily consumption. The list of drugs contained the following substances: nitrates, erectile dysfunction drugs, sedatives, cannabis, synthetic cannabinoids, “ecstasy” (MDMA, known in the context as “pills”), MDMA (methylenedioxymethamphetamine, known in the context as “crystal-amphetamine”, methamphetamine, mephedrone, and synthetic stimulants), GHB/GBL (gamma hydroxybutyrate/gamma butyrolactone), ketamine, LSD (lysergic acid diethylamide), cocaine powder, and cocaine base. In the present study, we clustered patterns of drug consumption by using latent class analysis (LCA), which is described in the statistical-analysis section.

We also screened for nicotine dependence, using the Fagerström test, which is a standard instrument for assessing the intensity of physical addiction to nicotine [42]. It comprises 6 items with 2 or more response options. The final score ranges from 0 to 10, with higher scores indicating a high degree of tobacco dependence. Given the suggested cutoff points and our research aims, a 3-level variable was created: non-smoker, low nicotine dependence (0–3), and medium–high nicotine dependence (4 or more).

We also collected data about sexual partners. We asked about sexual partners during the previous 6 months, coded as none (if the individual reported not having sexual activity), steady partner and occasional partner, only steady partner, and only occasional partners. We also recorded the number of sexual partners, and this was categorized into terciles according to the frequencies (tercile 1 [0–3], tercile 2 [3–7], and tercile 3 [7–360]).

Finally, the participants were asked if they had used any of the previously mentioned drugs with the intention of engaging in a long session of one-on-one sex, threesome, or group-sex party in a private home or in a commercial sex venue (sexualized drug use) once in a lifetime, in the previous year, or in the previous month. This variable was coded as no, once in a lifetime, last year, and last month.

##### Social Environment

We collected information about social issues. Regarding serodisclosure, participants were asked how many people in their environment knew they had HIV. The response was coded as none, less than half, and almost all or all. In terms of discrimination, 2 questions focused on the manner in which the participant was received in the health services, as follows: “Have you been treated differently at your health center?” and “Were you denied care or delayed treatment at your health center?” Possible answers were “never”, “rarely”, “sometimes”, and “often”, which were recoded into yes/no (“never”). Moreover, we administered the NEURO-QOL Item Bank v1.0—stigma and the NEURO-QOL Item Bank v1.1 satisfaction with social roles to evaluate self-perceived stigma and satisfaction with social roles, respectively [39]. In the case of self-perceived stigma, higher scores are related to a greater perception of stigma, while for satisfaction with social roles, higher scores mean greater satisfaction with social roles, as in the cognitive function measure.

Social isolation was estimated using the Patient-Reported Outcomes Measurement Information System (PROMIS) Item Bank v2.0—Social Isolation 8a, a universal platform with standardized and language-adapted measures, and common terminology and metrics that enable comparisons across domains and the population [43]. It refers to perceptions of being avoided, excluded, or unknown by others, without establishing a time frame. In terms of scoring, each question has 5 response options, scored from 1 to 5. The total raw score is obtained by adding the values of the responses for each item and then transforming them into T-scores, with a mean of 50 and a standard deviation of 10. In this construct, higher scores are indicative of a higher degree of social isolation.

Weekly hours dedicated to leisure activities and weekly hours spent caring for others were also collected.

### 2.3. Statistical Analyses

We used multiple imputation algorithms to manage missing data for all potential confounding and exposure variables among all participants, generating 20 imputed datasets (Appendix A).

To include illegal drug consumption profiles in our analyses, we conducted an LCA with the aim of categorizing an individual’s drug use during the previous 12 months. We included 14 recreational drugs—nitrates, phosphodiesterase-5 blockers and other erectile-dysfunction medication, natural or synthetic cannabinoids, amphetamines, methamphetamines, mephedrone or other synthetic stimulants (i.e., MDMA or “ecstasy”), GHB/GBL, ketamine, LSD, and cocaine—as observed indicators to identify classes for drug use. We ran the model from 1 to 10 latent classes and eventually chose the optimal number of latent classes after considering the following indicators: the lowest value of the adjusted Bayesian information criterion (aBIC), the consistent Akaike information criterion (CAIC), and the entropy index (values close to 0.80) and interpretability and clinical criteria [44].

We then performed a descriptive analysis stratifying by outcome (prevalence of depressive symptoms) and by prevalence of depressive symptoms and gender. We used measures of central tendency and dispersion for quantitative variables (median and interquartile range). For categorical variables, we calculated absolute frequencies and percentages. We also used the χ^2^ test and Mann–Whitney test to assess the association between each exposure independently and the outcome.

Finally, we fitted a multivariable logistic regression model. We used LASSO regression (Least Absolute Shrinkage and Selection Operator) as our variable selection model to avoid overfitting (Appendix A), considering the 20 imputed datasets. Continuous variables were included without further modifications. The odds ratio (OR) of continuous variables represents a change of 1 unit. We fixed gender as a potential confounding variable. We used Rubin’s rules to aggregate the results from the 20 imputed datasets [45]. The data were analyzed using R version 4.1.0 [46].

## 3. Results

### 3.1. General Characteristics of the Study Population by Prevalence of Depressive Symptoms

Of the total number of people invited to participate (n = 1092), 1060 (97%) completed the surveys. The main characteristics of the sample according to the prevalence of depressive symptoms are presented in Table 1. The sample mainly comprised men (n = 833, 78.58%), followed by women (n = 192, 18.11%) and transgender people (n = 35; 3.30%). As for age, 58.02% were aged 40–59 years, 22.26% aged ≤ 39 years, and 19.72% aged ≥ 60 years. Approximately half of the sample (51.32%) had higher education, 56.79% were currently working, and 40.09% reported having an income between EUR 1001 and EUR 2000 per month. Sex workers made up 3.58% of the sample, and no significant differences were found between people with and without depressive symptoms (*p* = 0.176).

With respect to comorbidities, we observed that participants with depressive symptoms presented worse scores in HRQoL, both in the physical dimension (Md = 54.9 [IQR = 39.44–71.14]) and in the mental dimension (Md = 68.01 [IQR = 50.5–78.33]) (*p* < 0.001), as well as worse cognitive function (Md = 41.60 [IQR = 26.28–57.01], *p* < 0.001). Most of the individuals in the sample (82.45%) reported being satisfied with their sex life, although a higher percentage of people with depressive symptoms were dissatisfied (33.92%) than people without depressive symptoms (13.09%) (*p* < 0.001). As for alcohol consumption, 26.60% were non-drinkers, and 18.30% were high-risk drinkers. In the case of people with depressive symptoms, these percentages were 35.68% and 17.18%, respectively.

As for HIV-related variables, only 7.26% reported that they had been diagnosed in the 12 months before the survey, with the most common mode of transmission being sex between men (MSM: 58.02%) and via drug injection (PWID: 20.00%).

Concerning health-related behaviors, drug use was categorized using an LCA based on the types of drugs taken during the previous 12 months. Three clusters of drug use were identified. Cluster 1 (77.64%) comprised mostly participants who did not consume drugs or consumed drugs such as cannabis, cocaine, erectile-dysfunction medication, or nitrates, with 4% polyconsumption of two drugs at most. In Cluster 2 (13.68%), the prevalence of drug consumption was >50% (cannabis, cocaine, or nitrates), a low prevalence of consumption of stimulants (MDMA, amphetamines, and methamphetamines), and polyconsumption of between two and six drugs. Cluster 3 (8.68%) contained patients with a high prevalence of stimulants, drugs used during sex (GHB, mephedrone, and erectile dysfunction drugs), and ketamine, and higher polyconsumption, i.e., between 4 and 13 drugs simultaneously (Appendix A). Regarding nicotine dependence, more than half of the people in the sample (57.26%) were non-smokers, followed by 22.92% of individuals with medium–high nicotine dependence and 19.81% with low nicotine dependence. The percentage of medium–high nicotine dependence was greater among participants with depressive symptoms (32.60%, *p* < 0.001). The percentage of people with depressive symptoms who reported not having sex was higher (28.19%) than that of people who did not have depressive symptoms (16.69%) (*p* < 0.001). Finally, 45.47% of the participants reported sexualized drug use.

As for social-environment-related determinants, we found that most of the people in the sample (65.57%) had revealed their serostatus to less than half of the people they knew, and 15.94% of the participants replied that nobody knew they had HIV. Concerning self-perceived stigma and discrimination in healthcare centers, 19.91% of the sample reported being treated differently, and 13.02% had been denied care or treatment, or their care or treatment had been delayed. Among people who had been treated differently in healthcare centers, there was a higher percentage of individuals with depressive symptoms (31.72%) than without depressive symptoms (16.69%) (*p* < 0.001). Similarly, stigma and discrimination scores were higher among participants with depressive symptoms (Md = 13, [IQR = 8.00–33.00] vs. Md = 10, [8.00–23.00], *p* < 0.001). Furthermore, perception of social isolation was greater among participants with depressive symptoms (Md = 55.10, [IQR = 34.00–69.40], *p* < 0.001), and satisfaction with social roles was lower than in those without symptoms (Md = 42.70 [IQR = 37.26–49.20] vs. Md = 41.40 [IQR = 34.00–58.46], Md = 48.00 [41.10–49.20], *p* < 0.001), respectively. To conclude, the mean weekly number of hours spent caring for others was higher in participants with depressive symptoms (mean 10.49 [SD = 23.19]) than in those without depressive symptoms (5.25 [SD = 13.20], *p* = 0.001).

### 3.2. General Characteristics of the Study Population by Gender and Prevalence of Depressive Symptoms

To explore gender differences between people with and without depressive symptoms, we stratified the sample by gender (men, women, and transgender; see Table 2). Regarding sociodemographic variables, men with depressive symptoms had a lower educational level (27.81%), were unemployed in a higher proportion (31.13%), and had a lower monthly income (17.88% were no income, and 47.02% were <EUR 1000) than those without depressive symptoms (*p*-values ≤ 0.001 for all of them). No statistically significant results were found among women and transgender people.

Regarding comorbidities and social-environment-related variables, we found that, regardless of gender, people with depressive symptoms had worse HRQoL, poorer cognitive function (although the lowest score was found among transgender people with depressive symptoms; Md = 39.00 [IQR = 32.53–50.83], *p*-value ≤ 0.001), a higher perception of social isolation, and lower satisfaction with their social role. The perception of stigma and discrimination was greater among men and women with depressive symptoms, (Md = 13, [IQR = 8.00–32.25] and Md = 13, [IQR = 8.00–34.00]), respectively, and was statistically significant when compared to their counterparts without depressive symptoms (Md = 10, [IQR = 8.00–21.98] and Md = 10, [IQR = 8.00–25.00], respectively, *p*-value ≤ 0.001).

### 3.3. Factors Associated with the Prevalence of Depressive Symptoms

Table 3 presents the results for the final model, which identified the main risk and protective factors for depressive symptoms. According to our model, comorbidities and social-environment-related variables were strongly associated with mental health. Depressive symptoms were more prevalent among women (OR = 1.34 [CI, 0.77–2.32]) and transgender people (OR = 1.26 [CI, 0.43–3.70]) than among men. They were also more prevalent among people with a lower monthly income, since a low income increased the risk of high levels of depressive symptoms, as follows: >EUR 2001 vs. no income, OR = 0.40 (CI, 0.14–1.11); EUR 1001–2000 vs. no income, OR = 0.53 (CI, 0.24–1.18); <EUR 1000 vs. no income, OR = 1.17 (CI, 0.56–2.44); however, these results were not significant. In addition, depressive symptoms were positively associated with social isolation (OR = 1.05 [CI, 1.02–1.08]), poor physical HRQoL (OR = 1.06 [CI, 1.02–1.09]), and mental HRQoL (OR = 1.13 [CI, 1.09–1.17]). On the contrary, depressive symptoms were negatively associated with satisfaction with social role (OR = 0.86 [CI, 0.79–0.94]), cognitive function (OR = 0.92 [CI, 0.89–0.95]), serodisclosure to all or most of people known (vs. none; OR = 0.39 [CI, 0.17–0.87]), and sexualized drug use once in a lifetime (vs. no; OR = 0.52 [CI, 0.29–0.93]).

## 4. Discussion

The present study sought to identify the determinants of depressive symptoms among PLWH who are receiving hospital care by examining group differences. We found that the overall prevalence of depressive symptoms was 21.4%. This is nearly one-quarter of PLWH, and the prevalence is higher among women and transgender people. PLWH with depressive symptoms had worse HRQoL, poorer cognitive function, lower satisfaction with social roles, a higher perception of social isolation, and a higher perception of stigma and discrimination. Despite having a life expectancy that is almost equivalent to that of people without HIV, PLWH continue to face unmet needs in terms of the burden of depressive symptoms and poor HRQoL [6].

PLWH experience high rates of depressive symptoms and related psychosocial risk factors that vary by gender [24]. Some studies have reported a greater prevalence of depressive symptoms in WLWH (also in terms of severity) [13,14], yet data are much more limited for transgender people, given that only their biological sex is considered, as opposed to their gender identity [47,48]. Our results confirm a higher prevalence of depressive symptoms in women and transgender people than in men. Furthermore, we found that women with depressive symptoms had poorer HRQoL, worse cognitive function, and a more marked perception of being discriminated against in healthcare centers than men with depressive symptoms. Transgender people with depressive symptoms showed a greater perception of social isolation and dissatisfaction with social roles than depressed men. Most of these results were similar to those found in previous research. For example, WLWH have been shown to have a significantly lower HRQoL than men with HIV [49,50]. As with depressive symptoms, poorer HRQoL in WLWH is a multi-causal phenomenon involving economic and social factors that limit or facilitate access to health resources and services [51]. Concerning perceived stigma and social isolation, previous evidence has shown that WLWH and transgender people tend to experience greater discrimination than men, and this translates into lower rates of social support, greater social isolation, and less frequent attendance at health services to avoid discrimination [52]. The fact that, in most cases, outcomes were worse for women and transgender people suggests a greater vulnerability that might translate into a higher risk of developing depression or other mental health problems. However, transgender people are under-represented in our sample, and this may have affected the significance of our results.

Social isolation has previously been identified as an underestimated risk factor for health and prevalence of depression and is common in PLWH, especially WLWH [53,54]. As might be expected, a poor social environment would lead to dissatisfaction with social roles, and this would trigger or worsen depressive symptoms and, in turn, lead to a poorer HRQoL. Furthermore, those who had disclosed their HIV status to all or almost all people around them had a lower prevalence of depressive symptoms. According to previous research, internalized stigma may play a role in disclosure of HIV status; that is, both stigma and depression were negatively correlated with serodisclosure and social connectedness [54,55].

Another interesting finding was that better cognitive function was associated with a lower risk of depressive symptoms. The fact is that cognitive impairment, especially in the domains of concentration, attention, processing speed, and working memory, is common among persons with depressive symptoms. However, in the case of PLWH, it is present in almost 50% of cases [56]. Results from a systematic review and meta-analysis concluded that cognitive impairment is a complication of the depressive symptoms, leading to poorer HRQoL, and that this impairment persists even though the person recovers from depression [57]. Moreover, both low mood and cognitive impairment are associated with poor psychosocial functioning [57]. Thus, preventing depressive symptoms takes on an even more relevant role given the persistent effects of cognitive impairment on patients with depressive symptoms, on psychosocial functioning, and on HRQoL.

It is worth mentioning the negative association found between sexualized drug use once in a lifetime and depressive symptoms. A recent systematic review found mixed results regarding the relationship between mental health and sexualized drug use [58]. An increased risk of depression, anxiety, and substance abuse has been reported in persons who engaged in sexualized drug use ([59,60]); however, some studies did not find this relationship [61,62]. Vaccher et al. (2020) found that men who had used poppers (nitrates), both in the past and more recently, obtained lower scores on depression and anxiety scales than men who had never used them [61]. Similarly, Hammoud et al. (2018) reported fewer symptoms of depression in men who had used GHB in the previous 6 months than in those who had not [62]. Otherwise, in the case of intravenous use of psychoactive substances in a sexual context (slamsex), a study conducted in Spain revealed that participants who engaged in slamsex were more likely to have depression, anxiety, and drug abuse disorders than those who engaged in non-injecting sexualized drug use [63]. These results suggest that this association may be influenced by the type of substance, as well as the frequency of use (i.e., once in a lifetime, in the previous year, or in the previous 6 months). In the present paper, we created a variable that took into account temporality criteria, although we did not perform specific drug analyses. Therefore, an in-depth analysis of each type of substance, polydrug use, and frequency of use would shed light on this phenomenon.

To conclude, we highlight those determinants of depressive symptoms that may be modifiable through strategic interventions, such as social isolation and the perception of stigma and discrimination. Both factors may be associated with serodisclosure to a higher number of people and better satisfaction with social roles, which, as a consequence, would lead to better HRQoL, including mental health. Addressing these aspects that consider gender diversity is even more important given gender inequalities (see above), especially when interventions are primarily designed for MSM. Multilevel system-strengthening approaches that integrate mental healthcare into HIV care and prevention within healthcare and community-based organizations need to be addressed in Spain and to be tailored to specific subgroups of PLWH that are at a higher risk of depressive symptoms.

### Strengths and Limitations

One of the strengths of the present work is that it includes a representative sample of PLWH with sufficient power to identify the variables in the study. Furthermore, we conducted an exhaustive assessment including several measures that might be related to depressive symptoms, such as comorbidities, health-related behaviors, and social-environment-related variables, without restricting our research to sociodemographic variables [5]. This provides a good approximation of the factors affecting mental health. In addition, the assessment instruments used have been validated in PLWH and lend greater rigor to our research. Finally, as mentioned earlier, we analyzed the data based on gender instead of biological sex, since gender inequalities play an important role in the factors associated with depressive symptoms in PLWH [26]. Embracing a gender approach is essential if we are to translate the needs of PLWH into effective clinical interventions [26].

Our findings are also subject to a series of limitations. First, interviews were conducted using an electronic tablet. Consequently, those unfamiliar with new technologies, such as older people, may have had some difficulty participating in the study. However, we consider that the presence of a peer during the interview overcame this possible barrier. Second, we used a convenience sample. The sample was not randomized, and specific strata were chosen, namely gender, age, and hospital of recruitment. Therefore, the sample can be considered representative of PLWH. In addition, we decided to oversample women and people aged 60 years and older, since these groups are often under-represented in HIV-related studies. Third, we used multiple imputations to avoid missing data in our database. While this might have introduced some bias, the imputed variables did not exceed more than 25% of missing data. Moreover, quality estimations were calculated to ensure that there were no statistically significant differences between the non-imputed and the imputed databases in terms of outcome. Fourth, since Vive+ was a cross-sectional study, causality could not be addressed. Most of the associations found may have a bidirectional relationship with mental health; therefore, a longitudinal study could shed light on the directionality of these risk and protective factors with respect to health and, consequently, HRQoL among PLWH. The Vive+ study is expected to conduct future follow-ups, thus making it a longitudinal study that will explore the effects of these variables on mental health and HRQoL. Fifth, we used a single-item question to measure gender. Although different categories were available (woman, man, trans woman, trans man, genderqueer, and other), this might be insufficient for the detection of trans people. Future studies should include categorical lists where the participant can select more than one option, as well as include an open-text option where people can report their gender identity [64].

## 5. Conclusions

Depressive symptoms are common among PLWH in general and are more common in women and transgender people than in men, since these populations face systematic structural barriers aggravated by the intersectionality of various factors (e.g., gender, socioeconomic status, HIV status, and social context). Furthermore, the effect of psychosocial variables, such as HRQoL, social isolation, and satisfaction with social roles, on the symptoms of depression highlights the multidimensionality of this problem and the need to create psychological care services. Among the variables significantly associated with depressive symptoms that could be modified, we identified social isolation and perceptions of stigma and discrimination, which could, in turn, facilitate serodisclosure and satisfaction with social roles. Appropriate psychological care, taking into account this intersectionality, could improve the mental health and HRQoL of the PLWH. Therefore, more studies are needed to assess barriers to mental health services among PLWH. Studies should lead to community-based interventions aimed at removing these barriers. Given the prevalence of clinically significant depressive symptoms, it may be important for healthcare providers to incorporate a gender perspective into their daily clinical practice.

## Figures and Tables

**Table 1 ijerph-20-03687-t001:** Characteristics of the total sample by prevalence of depressive symptoms.

Variables	Total Sample n = 1060	No Depressive Symptoms n = 833, 78.58%	Depressive Symptoms n = 227 ^a^, 21.42%	*p*-Value
** *Sociodemographic variables* **				
**Gender, n (%)**				<0.001
Men	833 (78.58)	682 (81.87)	151 (66.52)	
Women	192 (18.11)	129 (15.49)	63 (27.75)	
Transgender	35 (3.30)	22 (2.64)	13 (5.73)	
**Age group, n (%)**				0.792
≤39	236 (22.26)	188 (22.57)	48 (21.15)	
40–59	615 (58.02)	484 (58.10)	131 (57.71)	
≥60	209 (19.72)	161 (19.33)	48 (21.15)	
**Born abroad (yes), n (%)**	354 (33.40)	284 (34.09)	70 (30.84)	
**Level of education, n (%)**				<0.001
None or primary school	236 (22.26)	165 (19.81)	71 (31.28)	
Secondary school	280 (26.42)	214 (25.69)	66 (29.07)	
Higher education	544 (51.32)	454 (54.50)	90 (39.65)	
**Occupation, n (%)**				<0.001
Currently working	602 (56.79)	514 (61.7)	88 (38.77)	
Not currently working	168 (15.85)	111 (13.33)	57 (25.11)	
Retired	161 (15.19)	127 (15.25)	34 (14.98)	
Homemaker	25 (2.36)	17 (2.04)	8 (3.52)	
On leave	104 (9.81)	64 (7.68)	40 (17.62)	
**Income, n (%)**				<0.001
No income	97 (9.15)	64 (7.68)	33 (14.54)	
<EUR 1000	391 (36.89)	267 (32.05)	124 (54.63)	
EUR 1001–2000	425 (40.09)	370 (44.42)	55 (24.23)	
>EUR 2001	147 (13.87)	132 (15.85)	15 (6.61)	
**Sexual orientation, n (%)**				<0.001
Heterosexual	417 (39.34)	303 (36.37)	114 (50.22)	
Homosexual	530 (50.00)	443 (53.18)	87 (38.33)	
Bisexual	113 (10.66)	87 (10.44)	26 (11.45)	
**Recent sex work (yes), n (%)**	38 (3.58)	26 (3.12)	12 (5.29)	0.176
** *Comorbidities* **				
**HRQoL—physical, median [IQR]**	47.74 [37.11–66.42]	46.09 [36.82–63.98]	54.9 [39.44–71.14]	<0.001
**HRQoL—mental, median [IQR]**	56.73 [36.03–72.48]	53.80 [34.41–69.04]	68.01 [50.5–78.33]	<0.001
**Cognitive function, median [IQR]**	50.50 [32.65–64.20]	52.60 [38.90–64.20]	41.60 [26.28–57.01]	<0.001
**Overall satisfaction with sex life, n (%)**				<0.001
Satisfied	874 (82.45)	724 (86.91)	150 (66.08)	
Unsatisfied	186 (17.55)	109 (13.09)	77 (33.92)	
**Alcohol dependence, n (%)**				0.002
Non-drinker	282 (26.60)	201 (24.13)	81 (35.68)	
Low-risk drinker	584 (55.09)	477 (57.26)	107 (47.14)	
High-risk drinker	194 (18.30)	155 (18.61)	39 (17.18)	
**Diagnosed in the last 12 months (yes), n (%)**	77 (7.26)	61 (7.32)	16 (7.05)	1
**Mode of transmission, n (%)**				<0.001
PWID	212 (20.00)	144 (17.29)	68 (29.96)	
MSM	615 (58.02)	512 (61.46)	103 (45.37)	
HHTX	87 (8.21)	76 (9.12)	11 (4.85)	
MHTX	146 (13.78)	101 (12.12)	45 (19.82)	
** *Health-related behaviors* **				
**Substance use, n (%) ^b^**				0.42
Cluster 1	823 (77.64)	653 (78.39)	170 (74.89)	
Cluster 2	145 (13.68)	108 (12.97)	37 (16.30)	
Cluster 3	92 (8.68)	72 (8.64)	20 (8.81)	
**Nicotine dependence, n (%)**				<0.001
Non-smoker	607 (57.26)	492 (59.06)	115 (50.66)	
Low nicotine dependence	210 (19.81)	172 (20.65)	38 (16.74)	
Medium–high nicotine dependence	243 (22.92)	169 (20.29)	74 (32.60)	
**Sexual partners in the last 6 months, n (%)**				<0.001
None	203 (19.15)	139 (16.69)	64 (28.19)	
Steady partner and occasional partner	165 (15.57)	140 (16.81)	25 (11.01)	
Only steady partner	418 (39.43)	333 (39.98)	85 (37.44)	
Only occasional partners	274 (25.85)	221 (26.53)	53 (23.35)	
**Number of sexual partners, n (%)**				0.109
Tercile 1 [0–3]	183 (17.26)	152 (18.25)	31 (13.66)	
Tercile 2 [3–7]	115 (10.85)	94 (11.28)	21 (9.25)	
Tercile 3 [7–360]	141 (13.30)	115 (13.81)	26 (11.45)	
Not applicable	621 (58.58)	472 (56.66)	149 (65.64)	
**Sexualized drug use, n (%)**				0.168
No	578 (54.53)	461 (55.34)	117 (51.54)	
Once in a lifetime	236 (22.26)	189 (22.69)	47 (20.70)	
Last year	106 (10.00)	83 (9.96)	23 (10.13)	
Last month	140 (13.21)	100 (12.00)	40 (17.62)	
** *Social environment* **				
**Serodisclosure, n (%)**				0.409
None	169 (15.94)	138 (16.57)	31 (13.66)	
Less than half	695 (65.57)	538 (64.59)	157 (69.16)	
All or almost all	196 (18.49)	157 (18.85)	39 (17.18)	
**At your health center: had been treated differently (yes), n (%)**	211 (19.91)	139 (16.69)	72 (31.72)	<0.001
**At your health center: treatment/care denied or delayed (yes), n (%)**	138 (13.02)	59 (25.99)	79 (9.48)	<0.001
**Stigma and discrimination, median [IQR]**	10 [8.00–26.00]	10 [8.00–23.00]	13 [8.00–33.00]	<0.001
**Social isolation, median [IQR]**	43.20 [34.00–64.65]	41.40 [34.00–58.46]	55.10 [34.00–69.40]	<0.001
**Satisfaction with social roles, median [IQR]**	47.20 [38.90–49.20]	48.00 [41.10–49.20)	42.70 [37.26–49.20]	<0.001
**Weekly hours dedicated to leisure, mean (SD)**	15.75 (16.30)	16.20 (16.01)	14.12 (17.23)	0.103
**Weekly hours spent caring for others, mean (SD)**	6.37 (16.01)	5.25 (13.20)	10.49 (23.19)	0.001

Note: IQR, interquartile range; SD, standard deviation; PWID, people who inject drugs; MSM, men who have sex with men; HHTX, men who have sex with women; MHTX, women who have sex with men; HRQoL, health-related quality of life. ^a^ Confidence intervals [CIs] for depressive symptoms: [CI, 19.01–23.98]. ^b^ Cluster 1 (77.64%): participants who did not consume drugs, or consumed cannabis, cocaine, erectile dysfunction medication, or nitrates, with 4% polyconsumption of 2 drugs at most in the previous year. Cluster 2 (13.68%): prevalence >50% of cannabis, cocaine, or nitrates; a low prevalence of consumption of stimulants (MDMA, amphetamines, and methamphetamines); and polyconsumption of between 2 and 6 drugs. Cluster 3 (8.68%): high prevalence of common stimulants, drugs used during sex (GHB, mephedrone, erectile dysfunction drugs), and ketamine and higher polyconsumption (between 4 and 13 drugs during the previous year).

**Table 2 ijerph-20-03687-t002:** Characteristics of the total sample by gender and prevalence of depressive symptoms.

Variables	Gender								
	Men n = 827			Women n = 190			Transgender n = 35		
	No Depressive Symptoms n = 682, 81.87%	Depressive Symptoms ^a^ n = 151, 18.13%	*p*-Value	No Depressive Symptoms n = 129, 67.19%	Depressive Symptoms ^a^ n = 63, 32.81%	*p*-Value	No Depressive Symptoms n = 22, 62.86%	Depressive Symptoms ^a^ n = 13, 37.14%	*p*-Value
** *Sociodemographic variables* **									
**Age, n (%)**			0.992			0.789			0.867
<39	166 (24.34)	36 (23.84)		17 (13.18)	8 (12.70)		5 (22.73)	4 (30.77)	
40–59	390 (57.18)	87 (57.62)		83 (64.34)	38 (60.32)		11 (50.00)	6 (46.15)	
>60	126 (18.48)	28 (18.54)		29 (22.48)	17 (26.98)		6 (27.27)	3 (23.08)	
**Born abroad (yes), n (%)**	238 (34.90)	50 (33.11)	0.747	35 (27.13)	11 (17.46)	0.196	11 (50.00)	9 (69.23)	0.449
**Level of education, n (%)**			<0.001			0.686			0.667
None or primary school	113 (16.57)	42 (27.81)		48 (37.21)	25 (39.68)		4 (18.18)	4 (30.77)	
Secondary school	169 (24.78)	47 (31.13)		34 (26.36)	13 (20.63)		11 (50.00)	6 (46.15)	
Higher education	400 (58.65)	62 (41.06)		47 (36.43)	25 (39.68)		7 (31.82)	3 (23.08)	
**Occupation, n (%)**			<0.001			0.132			0.905
Currently working	447 (65.54)	61 (40.40)		56 (43.41)	20 (31.75)		11 (50)	7 (53.85)	
Not currently working	84 (12.32)	47 (31.13)		22 (17.05)	8 (12.70)		5 (22.73)	2 (15.38)	
Retired	104 (15.25)	18 (11.92)		19 (14.73)	14 (22.22)		4 (18.18)	2 (15.38)	
Homemaker	1 (0.15)	2 (1.32)		16 (12.40)	6 (9.52)		0 (0.00)	0 (0.00)	
On leave	46 (6.74)	23 (15.23)		16 (12.40)	15 (23.81)		2 (9.09)	2 (15.38)	
**Income, n (%)**			<0.001			0.098			0.230
No income	39 (5.72)	27 (17.88)		23 (17.83)	5 (7.94)		2 (9.09)	1 (7.69)	
<EUR 1000	192 (28.15)	71 (47.02)		65 (50.39)	43 (68.25)		10 (45.45)	10 (76.92)	
EUR 1001–2000	327 (47.95)	40 (26.49)		37 (28.68)	13 (20.63)		6 (27.27)	2 (15.38)	
>EUR 2001	124 (18.18)	13 (8.61)		4 (3.10)	2 (3.17)		4 (18.18)	0 (0.00)	
**Sexual orientation, n (%)**			0.137			0.56			0.833
Heterosexual	180 (26.39)	50 (33.11)		116 (89.92)	59 (93.65)		7 (31.82)	5 (38.46)	
Homosexual	434 (63.64)	83 (54.97)		0 (0.00)	0 (0.00)		9 (40.91)	4 (30.77)	
Bisexual	68 (9.97)	18(11.92)		13 (10.08)	4 (6.35)		6 (27.27)	4 (30.77)	
**Recent sex work (yes), n (%)**	20 (2.93)	9 (5.96)	0.112	1 (0.78)	1 (1.59)	1	5 (22.73)	2 (15.38)	0.930
** *Comorbidities* **									
**HRQoL—physical, median [IQR]**	45.50 [36.84–63.21]	53.27 [39.24–70.75]	<0.001	49.48 [37.39–64.76]	56.47 [42.15–72.23]	<0.001	53.92 [37.72–64.40]	58.05 [41.19–69.97]	0.282
**HRQoL—mental, median [IQR]**	53.59 [34.41–68.99]	68.08 [50.53–78.32]	<0.001	54.49 [34.41–69.44]	68.21 [52.91–74.97]	<0.001	54.77 [38.39–68.10]	65.62 [57.33–75.96]	0.001
**Cognitive function, median [IQR]**	52.60 [38.90–64.20]	41.70 [26.38–55.67]	<0.001	52.60 [39.06–64.20]	42.20 [26.96–64.20]	<0.001	49.65 [38.72–64.20]	39.00 [32.53–50.89]	<0.001
**Overall satisfaction with sex life, n (%)**			<0.001			0.005			0.930
Satisfied	600 (87.98)	99 (65.56)		107 (82.95)	40 (63.49)		17 (77.27)	11 (84.62)	
Unsatisfied	82 (12.02)	52 (34.44)		22 (17.05)	23 (36.51)		5 (22.73)	2 (15.38)	
**Alcohol dependence, n (%)**			0.041			0.444			0.817
Non-drinker	144 (21.11)	46 (30.46)		49 (37.98)	30 (47.62)		8 (36.36)	5 (38.46)	
Low-risk drinker	407 (59.68)	77 (50.99)		58 (44.96)	24 (38.10)		12 (54.55)	6 (46.15)	
High-risk drinker	131 (19.21)	28 (18.54)		22 (17.05)	9 (14.29)		2 (9.09)	2 (15.38)	
**Diagnosed in the last 12 months (yes), n (%)**	51 (7.48)	14 (9.27)	0.565	8 (6.20)	2 (3.17)	0.589	2 (9.09)	0 (0.00)	0.714
**Mode of transmission, n (%)**			<0.001			0.702			0.09
PWID	111 (16.28)	47 (31.13)		32 (24.81)	18 (28.57)		1 (4.55)	3 (23.08)	
MSM	495 (72.58)	93 (61.59)		0 (0.00)	0 (0.00)		17 (77.27)	10 (76.92)	
HHTX	76 (11.14)	11 (7.28)		0 (0.00)	0 (0.00)		0 (0.00)	0 (0.00)	
MHTX	0 (0.00)	0 (0.00)		97 (75.19)	45 (71.43)		4 (18.18)	0 (0.00)	
** *Health-related behaviors* **									
**Substance use, n (%) ^b^**			0.166			0.337			0.330
Cluster 1	515 (75.51)	103 (68.21)		122 (94.57)	57 (90.48)		16 (72.73)	10 (76.92)	
Cluster 2	96 (14.08)	29 (19.21)		6 (4.65)	6 (9.52)		6 (27.27)	2 (15.38)	
Cluster 3	71 (10.41)	19 (12.58)		1 (0.78)	0 (0.00)		0 (0.00)	1 (7.69)	
**Nicotine dependence, n (%)**			0.027			0.066			0.104
Non-smoker	402 (58.94)	75 (49.67)		76 (58.91)	34 (53.97)		14 (63.64)	6 (46.15)	
Low nicotine dependence	143 (20.97)	31 (20.53)		26 (20.16)	7 (11.11)		3 (13.64)	0 (0.00)	
Medium–high nicotine dependence	137 (20.09)	45 (29.80)		27 (20.93)	22 (34.92)		5 (22.73)	7 (53.85)	
**Sexual partners during the previous 6 months, n (%)**			0.144			0.310			0.294
None	89 (13.05)	30 (19.87)		44 (34.11)	29 (46.03)		6 (27.27)	5 (38.46)	
Steady partner and occasional partner	133 (19.50)	23 (15.23)		2 (1.55)	2 (3.17)		5 (22.73)	0 (0.00)	
Only steady partner	260 (38.12)	54 (35.76)		70 (54.26)	28 (44.44)		3 (13.64)	3 (23.08)	
Only occasional partners	200 (29.33)	44 (29.14)		13 (10.08)	4 (6.35)		8 (36.36)	5 (38.46)	
**Number of sexual partners, n (%)**			0.759			0.814			0.401
Tercile 1 [0–3]	349 (54.17)	84 (55.63)		114 (88.37)	57 (90.48)		9 (40.91)	8 (61.54)	
Tercile 2 [3–7]	134 (19.65)	25 (16.56)		11 (8.53)	5 (7.94)		7 (31.82)	1 (7.69)	
Tercile 3 [7–360]	89 (13.05)	19 (12.58)		4 (3.10)	1 (1.59)		1 (4.55)	1 (7.69)	
Not applicable	110 (16.13)	23 (15.23)		0 (0.00)	0 (0.00)		5 (22.73)	3 (23.08)	
**Sexualized drug use, n (%)**			0.080			0.102			0.744
No	341 (50.00)	66 (43.71)		109 (84.50)	45 (71.43)		11 (50.00)	6 (46.15)	
Once in a lifetime	170 (24.93)	33 (21.85)		16 (12.40)	12 (19.05)		3 (13.64)	2 (15.38)	
Last year	76 (11.14)	19 (12.58)		3 (2.33)	3 (4.76)		4 (18.18)	1 (7.69)	
Last month	95 (13.93)	33 (21.85)		1 (0.78)	3 (4.76)		4 (18.18)	4 (30.77)	
** *Social environment* **									
**Serodisclosure, n (%)**			0.517			0.739			0.562
None	111 (16.28)	19 (12.58)		21 (16.28)	10 (15.87)		6 (27.27)	2 (15.38)	
Less than half	441 (64.66)	103 (68.21)		84 (65.12)	44 (69.84)		13 (59.09)	10 (76.92)	
All or almost all	130 (19.06)	29 (19.21)		24 (18.60)	9 (14.29)		3 (13.64)	1 (7.69)	
**At your health center: had been treated differently (yes), n (%)**	105 (15.40)	42 (27.81)	<0.001	34 (26.36)	27 (42.86)	0.032	0 (0.00)	3 (23.08)	0.083
**At your health center:** **treatment/care denied or delayed (yes), n (%)**	63 (9.24)	35 (23.18)	<0.001	16 (12.40)	21 (33.33)	0.001	0 (0.00)	3 (23.08)	0.083
**Stigma and discrimination, median [IQR]**	10 [8.00–21.98]	13 [8.00–32.25]	<0.001	10 [8.00–25.00]	13 [8.00–34.00]	0.002	8 [8.00–28.32]	15 [8.00–22.00]	0.081
**Social isolation, median [IQR]**	41.50 [34.00–58.40]	55.40 [34.00–70.20]	<0.001	40.00 [34.00–55.62]	52.5 [34.00–68.64]	<0.001	34.00 [34.00–61.74]	57.50 [38.23–67.68]	<0.001
**Satisfaction with social roles, median [IQR]**	48 [41.10–49.20]	42.4 [36.95–48.52]	<0.001	47.80 [40.68–49.20]	43.40 [37.46–49.20]	<0.001	46.65 [41.85–49.20]	41.70 [38.16–48.93]	0.007
**Weekly hours dedicated to leisure, mean (SD)**	16.23 (13.73)	15.54 (18.59)	0.713	13.96 (17.81)	14.32 (15.43)	0.901	21.67 (21.27)	12.30 (19.18)	0.248
**Weekly hours spent caring for others, mean (SD)**	3.85 (10.07)	7.06 (15.48)	0.034	11.85 (22.83)	13.88 (27.85)	0.655	8.79 (20.35)	26.56 (57.20)	0.389

Note: IQR, interquartile range; SD, standard deviation; PWID, people who inject drugs; MSM, men who have sex with men; HHTX, men who have sex with women; MHTX, women who have sex with men; HRQoL, health-related quality of life. ^a^ Confidence intervals [CIs] for depressive symptoms: men [CI, 15.65–20.88], women [CI, 26.56–39.73], and transgender [CI, 23.16–53.66]. ^b^ Cluster 1 (77.64%): participants who did not consume drugs or consumed cannabis, cocaine, erectile dysfunction medication, or nitrates, with 4% polyconsumption of 2 drugs at most. Cluster 2 (13.68%): prevalence >50% of cannabis, cocaine, or nitrates; a low prevalence consumption of stimulants (MDMA, amphetamines, and methamphetamines); and polyconsumption of between 2 and 6 drugs. Cluster 3 (8.68%): high prevalence of common stimulants, drugs used during sex (GHB, mephedrone, and erectile-dysfunction drugs), and ketamine and higher polyconsumption (between 4 to 13 drugs during the previous year).

**Table 3 ijerph-20-03687-t003:** Final model from the regression analyses, with depression symptoms as the outcome variable in the total sample.

Variables	Coefficient	OR	95% CI	*p*-Value
**Intercept**	−3.875	0.02	[0.00–14.24]	0.245
**Gender**				
Men (ref.)				
Women	0.293	1.34	[0.77–2.32]	0.296
Transgender	0.237	1.26	[0.43–3.70]	0.664
**Income**				
No income (ref.)				
<EUR 1000	0.159	1.17	[0.56–2.44]	0.672
EUR 1001–2000	−0.620	0.53	[0.24–1.18]	0.123
>EUR 2001	−0.907	0.40	[0.14–1.11]	0.08
**Social isolation**	0.057	1.05	[1.02–1.08]	**<0.001**
**Satisfaction with social roles**	−0.141	0.86	[0.79–0.94]	**0.001**
**HRQoL—physical**	0.059	1.06	[1.02–1.09]	**<0.001**
**HRQoL—mental**	0.127	1.13	[1.09–1.17]	**<0.001**
**Cognitive function**	−0.079	0.92	[0.89–0.95]	**<0.001**
**Serodisclosure**				
None (ref.)				
Less than half	−0.445	0.64	[0.33–1.22]	0.181
All or almost all	−0.929	0.39	[0.17–0.87]	**0.023**
**Sexualized drug use**				
No (ref.)				
Once in a lifetime	−0.645	0.52	[0.29–0.93]	**0.027**
Last year	−0.239	0.787	[0.365–1.698]	0.541
Last month	−0.087	0.916	[0.457–1.838]	0.805

Note. HRQoL, health-related quality of life. Models were adjusted for gender and were run in 20 imputed datasets. Boldface indicates statistically significant results.

## Data Availability

The study protocol is available from Juliana Reyes-Urueña (e-mail: jmreyes@iconcologia.net). The statistical code for the analysis can be requested from Yesika Díaz, Sergio Moreno, and Jordi Aceiton (ydiazr@iconcologia.net, smorenof@iconcologia.net, jaceiton@igtp.cat). The data for this study are available at the Centre for Epidemiological Studies of Sexually Transmitted Diseases and HIV/AIDS in Catalonia (CEEISCAT), from the coordinating center of the PISCIS cohort, and from each of the collaborating hospitals upon request via https://pisciscohort.org/contacte/ (accessed on 15 February 2023).

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
