# Peer review of "Determinants of Depressive Symptoms in People Living with HIV: Findings from a Population-Based Study with a Gender Perspective"

_ijerph, 2023, doi:10.3390/ijerph20043687_

Round 1
Reviewer 1 Report
Thanks for working on this kind of project. Analyses of mental health among HIV people, especially between women and transgender are not common. I like your manuscript in general, but I think it could be improved following some recommendations.
There are lots of results, maybe separate them by category and with a specific title could help to read them better. I identified three parts: "General characteristics of study population by depression status", "General characteristics by gender and depression status" and "Associations with depression"
Would you try to run multivariate models by gender? I think to see the association of social and health variables separatley (especially for women, because the small size of transgender) could help to analyze if there is any difference between men and women. The results of the abstract are focused on the model with complete sample, and even if you include gender as covariate the specific associations by gender could be interesting. Moreover because the aim of the study is having a perspective of gender.
I think it could be also valuable to explain how the continuous variables such as “social isolation” and “satisfaction social role” were included in the multivariate model. In the descriptive section they were summarized with a median score, but I´m not sure if is the same in the multivariate logistic model. The ORs for these variables represent a change in the score of 1 unit? Is this really significantly different in terms of a social characteristic? I prefer keeping the continuous variables as continuous in a multivariate model, but I´m not sure if having 48 in social isolation instrument is really different of having 41. Is it?
In the sentences 298-301, the result of “poor satisfaction role” in the depressive group is a kind of unclear, maybe because the interpretation in opposite way to the previous social isolation degree. Could you please try to rewording to have an easier reading?
Results from table 3, show non-significant OR for sex (women and transgender) compared to men, but the sentence implies other thing, could you please rephrase it?
Minor details:
1.Related to explanation of the variables: “hours dedicated to leisure” and “hours for caring for others” are not clear. These are hours by week? By month? This is not specified neither in the text or in the tables.
2.Sentence 353, I think you mean “positively associated with” , it reads “positive associated with”.
3. The font text at end of the tables looks similar to the text in the results for next section and is not easy to see where the results text start (line 346)
4. Discussion, lines 377-381, you mean women with depressive symptoms compared to women without depressive symptoms have worse quality of life? Or you mean comparing with men with and without depressive symptoms? This result come from univariate comparison and the numbers looks very similar.
5.Lines 464-465, which variables had more than 25% of missings? If you don´t make imputation for those, what do you do?
Author Response
REVIEWER 1
Comments and Suggestions for Authors
Thanks for working on this kind of project. Analyses of mental health among HIV people, especially between women and transgender are not common. I like your manuscript in general, but I think it could be improved following some recommendations.
POINT 1: There are lots of results, maybe separate them by category and with a specific title could help to read them better. I identified three parts: "General characteristics of study population by depression status", "General characteristics by gender and depression status" and "Associations with depression"
RESPONSE 1: We are grateful to the reviewer for this suggestion and have included the following sections in the results section:
- General characteristics of the study population by prevalence of depressive symptoms
- General characteristics of the study population by gender and prevalence of depressive symptoms
- Factors associated with the prevalence of depressive symptoms
POINT 2: Would you try to run multivariate models by gender? I think to see the association of social and health variables separately (especially for women, because the small size of transgender) could help to analyze if there is any difference between men and women. The results of the abstract are focused on the model with complete sample, and even if you include gender as covariate the specific associations by gender could be interesting. Moreover because the aim of the study is having a perspective of gender.
RESPONSE 2: We agree that running multivariate models by gender is interesting and we did, in fact, consider such an analysis when we were working on the article. However, we dismissed it because of sample size constraints. The small sample of women (N=192) and transgender people (N=35) did not have sufficient statistical power to fit a reliable multivariate model.
In order to verify the reviewer's comments, we have re-run the analyses, considering separate models for women and for men. However, since these results do not provide additional information relevant to the objectives of the article, we have not included them. The results of these models are provided below for the reviewer's reference (see the attached file).
POINT 3: I think it could be also valuable to explain how the continuous variables such as “social isolation” and “satisfaction social role” were included in the multivariate model. In the descriptive section they were summarized with a median score, but I´m not sure if is the same in the multivariate logistic model. The ORs for these variables represent a change in the score of 1 unit? Is this really significantly different in terms of a social characteristic? I prefer keeping the continuous variables as continuous in a multivariate model, but I´m not sure if having 48 in social isolation instrument is really different of having 41. Is it?
RESPONSE 3: We have now included a sentence to clarify that the continuous variables were introduced without modification, i.e., as described in the methods section. Moreover, the odds ratios represented a change of one unit.
“Finally, we fitted a multivariable logistic regression model. We used LASSO regression (Least Absolute Shrinkage and Selection Operator) as our variable selection model to avoid over-fitting [Table S3], considering the 20 imputed datasets. Continuous variables were included without further modifications. The odds ratio (OR) of continuous variables in Table 3 represents a change of 1 unit. We fixed gender as a potential confounding variable. We used Rubin’s rules to aggregate the results from the twenty imputed datasets [42]. The data were analysed using R version 4.1.0 [43].”
POINT 4: In the sentences 298-301, the result of “poor satisfaction role” in the depressive group is a kind of unclear, maybe because the interpretation in opposite way to the previous social isolation degree. Could you please try to rewording to have an easier reading?
RESPONSE 4: In the case of social isolation, higher scores are indicative of greater perceived social isolation, whereas in satisfaction with social roles, higher scores represent greater satisfaction with social roles. We have rephrased the sentence as follows:
“Furthermore, perception of social isolation was higher among participants with depressive symptoms (Md=55.10, [IQR=34.00-69.40], p<0.001), and satisfaction with social roles was lower than in those without symptoms (Md=42.70 [IQR=37.26-49.20] vs. Md=41.40 [IQR=34.00-58.46], Md=48.00 [41.10-49.20], p<0.001, respectively).”
POINT 5: Results from table 3, show non-significant OR for sex (women and transgender) compared to men, but the sentence implies other thing, could you please rephrase it?
RESPONSE 5: We have modified this explanation slightly to make it clear that these results were not statistically significant.
“Depressive symptoms were more prevalent among women OR=1.34 [CI, 0.77-2.32] and transgender people (OR= 1.26 [CI, 0.43-3.70]) than among men. They were also more prevalent among people with lower monthly income, as low income increased the risk of high levels of depressive symptoms; >€2001 vs. no income, OR=0.40 [CI, 0.14-1.11]; €1001-2000 vs. no income, OR=0.53 [CI, 0.24-1.18]; <€1000 vs. no income, OR=1.17 [CI, 0.56-2.44]); however, these results were not significant.”
Minor details:
POINT 6: 1.Related to explanation of the variables: “hours dedicated to leisure” and “hours for caring for others” are not clear. These are hours by week? By month? This is not specified neither in the text or in the tables.
RESPONSE 6: We now state that these variables are expressed as hours per week spent on leisure and caring for others.
POINT 7: 2.Sentence 353, I think you mean “positively associated with” , it reads “positive associated with”.
RESPONSE 7: We have corrected this typo.
POINT 8: 3. The font text at end of the tables looks similar to the text in the results for next section and is not easy to see where the results text start (line 346)
RESPONSE 8: The journal template seems to have several formatting errors (the table captions are not in the correct font size, in some cases table captions are joined with the text that follows, tables 1 and 2 should be horizontal and in the same font as the rest of the text), and the reviewers have not had access to the supplementary material.
We have inserted comments to address these issues in the revised version and have contacted the journal.
POINT 9: 4. Discussion, lines 377-381, you mean women with depressive symptoms compared to women without depressive symptoms have worse quality of life? Or you mean comparing with men with and without depressive symptoms? This result come from univariate comparison and the numbers looks very similar.
RESPONSE 9: These results are presented in comparison to men with depressive symptoms. The text has been modified accordingly, as follows:
“Furthermore, we found that women with depressive symptoms had poorer HRQoL, worse cognitive function, and a more marked perception of being discriminated against in healthcare centres than men with depressive symptoms. Transgender people with depressive symptoms showed a greater perception of social isolation and dissatisfaction with social roles than depressed men.”
POINT 10: 5.Lines 464-465, which variables had more than 25% of missings? If you don´t make imputation for those, what do you do?
RESPONSE 10: Both reviewers' comments indicate that the supplementary materials were not provided by the journal during the review process. As explained in the statistical analysis section and detailed in the supplementary material, we performed a multiple imputation procedure to deal with missing data in our database. A detailed report on the proportion of missing data in each variable and the relevant analyses to ensure the quality of the imputations can be found in the supplementary material.
We agree that our explanation in the discussion section could be somewhat confusing and have modified the relevant text accordingly, as follows:
“Third, we used multiple imputation to avoid missing data in our database. While this might have introduced some bias, the imputed variables did not exceed more than 25% of missing data. Moreover, quality estimations were calculated to ensure that there were no statistically significant differences between the non-imputed and the imputed databases in terms of outcome.”

Reviewer 2 Report
Thank you for the opportunity to review this paper that present novel findings on the association of various risk and protective factors and depression among PLHV in Spain. As an under-explored topic in Spain, my strongest desire would be that the authors to weave in information on local Spain context (e.g., status on PLHV, healthcare availability, and policy for PLHV) throughout the paper. The paper can also benefit from substantial English editing. Below are some of my suggestions for the authors to consider.
Abstract
Line 29: This sentence can be interpreted as blaming PLWH for commonly having depressive symptoms. Perhaps reframe it as ‘People living with HIV (PLWH) are susceptible to a risk of developing depressive symptoms.’
Line 30: It’s usually not recommended to begin a sentence with number in an academic article. Perhaps rewrite as: PLWH (n = 1,060)
Line 34: This sentence requires a revision.
Line: 35: Replace ‘moreover…were associated’ with ‘we found…to associate’
Line 43: It will be great if the authors can insert another sentence here to illustrate the implication of their findings in Spain. For example, how can their findings be used to inform policy changes.
Introduction
Line 47-48: Consider breaking this long sentence into two for clarity purpose.
Line 55: Replace ‘bad’ with ‘poor’.
Line 58: This sentence needs revising as part on ‘often including non-representative samples’ doesn’t flow well with the former part.
Line 64: ‘What’ becomes an even more relevant intervention target? I also find this sentence difficult to comprehend as the authors combined multiple (yet different) individual and community factors together without an elaboration.
Line 67: ‘What’ is this population?
Line 70-74: The use of comma and semicolon interchangeably in this sentence is confusing.
Line 82: I wonder if the author meant ‘literature on LGBT+’ here?
Line 86: The term ‘gender perspective’ is problematic here (as well as in a few other locations) as most studies would have provided at least a ‘perspective’ on gender issues by silencing, marginalising, empowering and/or pathologising through cisgenderism. I suggest the authors to use a clearer term to clarify their intention, e.g., examining gender group differences?
Line 86: I suggest the authors to insert a paragraph to describe the status of PLHV in Spain and the type of health care (with a focus on mental healthcare) that this population can access. This can help to ground the paper within the Spain context.
Line 87: Can the authors name all the risk and protective factors that they assessed here?
Outcome
Line 126: The authors need to provide a Cronbach alpha for inter-reliability analysis of PHQ-9. Additional information on how the cut-off points are determined should be provided.
Predictor
Line 129: Information on how transgender identities are determined needs to be provided.
Statistical analysis:
Line 227: Tables S1, S2, and S3 can’t be found within the paper and are not accessible to reviewers.
Line 227: Information on missing data (e.g., percentage of missing for each variable) should be made transparent for readers.
Line 241: What were the qualitative variables assessed? Qualitative data indicate responses involving text.
Discussion
Line 441: It will be great if the authors can insert a paragraph here to discuss the implication of their findings for policy makers in Spain.
Author Response
REVIEWER 2:
Thank you for the opportunity to review this paper that present novel findings on the association of various risk and protective factors and depression among PLHV in Spain.
POINT 1: As an under-explored topic in Spain, my strongest desire would be that the authors to weave in information on local Spain context (e.g., status on PLHV, healthcare availability, and policy for PLHV) throughout the paper.
RESPONSE 1: We fully agree with the reviewer’s comment and no begin the introduction section as follows:
“In 2021, between 160,000 people were estimated to be living with HIV in Spain; the estimated prevalence of HIV in persons aged 15 to 49 years (0.3%) was similar to that reported in other European countries with concentrated epidemics, such as France (0.3%) and Italy (0.3%) [1]. At the end of 2019, it was estimated that 87% of people living with HIV (PLWH) were diagnosed, 97.3% of those diagnosed were receiving ART, and viral load was suppressed in 90.4% of those receiving ART [2]. Access to the Spanish healthcare system is universal and free of charge for all citizens. Individuals not born in Spain are also entitled to healthcare under the same conditions as Spanish citizens. In Spain, treatment for HIV is provided exclusively through hospital pharmacies; therefore, most HIV-infected patients receive HIV care and treatment in public hospitals [3]. HIV patients attend the HIV clinic for routine check-ups every 6 months if the infection is under control. Mental healthcare is included as part of routine follow-up.”
POINT 2: The paper can also benefit from substantial English editing. Below are some of my suggestions for the authors to consider.
RESPONSE 2: We have corrected the aspects mentioned by the reviewer and have had the manuscript proofread by a native-speaking medical writer to improve the written expression.
Abstract
- Line 29: This sentence can be interpreted as blaming PLWH for commonly having depressive symptoms. Perhaps reframe it as ‘People living with HIV (PLWH) are susceptible to a risk of developing depressive symptoms.’
- Line 30: It’s usually not recommended to begin a sentence with number in an academic article. Perhaps rewrite as: PLWH (n = 1,060)
- Line 34: This sentence requires a revision.
- Line: 35: Replace ‘moreover…were associated’ with ‘we found…to associate’
Introduction
- Line 47-48: Consider breaking this long sentence into two for clarity purpose.
- Line 55: Replace ‘bad’ with ‘poor’.
- Line 58: This sentence needs revising as part on ‘often including non-representative samples’ doesn’t flow well with the former part.
- Line 64: ‘What’ becomes an even more relevant intervention target? I also find this sentence difficult to comprehend as the authors combined multiple (yet different) individual and community factors together without an elaboration.
- Line 70-74: The use of comma and semicolon interchangeably in this sentence is confusing.
- Line 82: I wonder if the author meant ‘literature on LGBT+’ here?
- Line 67: ‘What’ is this population?
POINT 3: Line 43: It will be great if the authors can insert another sentence here to illustrate the implication of their findings in Spain. For example, how can their findings be used to inform policy changes.
RESPONSE 3: Thank you for this suggestion. The following sentence has been added at the end of the abstract.
“This study found that management of mental health issues is an area that needs to be improved and tailored to specific groups with the aim of enhancing the well-being of PLWH.”
POINT 4: Line 86: The term ‘gender perspective’ is problematic here (as well as in a few other locations) as most studies would have provided at least a ‘perspective’ on gender issues by silencing, marginalising, empowering and/or pathologising through cisgenderism. I suggest the authors to use a clearer term to clarify their intention, e.g., examining gender group differences?
RESPONSE 4: We agree with the reviewer that in some studies the term “gender perspective” may not be appropriate and have adjusted the text accordingly. We would be happy to make any further changes in this respect if the reviewer feels that they are necessary.
POINT 5: Line 86: I suggest the authors to insert a paragraph to describe the status of PLHV in Spain and the type of health care (with a focus on mental healthcare) that this population can access. This can help to ground the paper within the Spain context.
RESPONSE 5: This has been addressed in the response to point 1.
POINT 6: Line 87: Can the authors name all the risk and protective factors that they assessed here?
RESPONSE 6: We have included the groups of factors included in our analyses:
“The objective of the present study was to identify risk and protective factors for sociodemographic variables, comorbidities, health-related behaviours, and social environment–related variables associated with depressive symptoms in PLWH from a gender perspective in order to enable clinical and public health interventions.”
POINT 7: Outcome - Line 126: The authors need to provide a Cronbach alpha for inter-reliability analysis of PHQ-9. Additional information on how the cut-off points are determined should be provided.
RESPONSE 7: We have included Cronbach's alpha in the methods section when describing the outcome variable. Moreover, we included the bibliographic reference from which we extracted the cut-off points.
POINT 8: Predictor - Line 129: Information on how transgender identities are determined needs to be provided.
RESPONSE 8: In the Vive+ study, participants were asked about their gender. We included the following terms: man, woman, transgender woman, transgender man, genderqueer, and additional gender category. No participant selected the genderqueer or the additional gender category. As stated in the description in the “Variables” section, we combined transgender men and women owing to the small sample sizes (N=9, and N=26, respectively). We now state that we refer to self-reported gender.
POINT 9: Statistical analysis - Line 227: Tables S1, S2, and S3 can’t be found within the paper and are not accessible to reviewers.
RESPONSE 9: Both reviewers' comments lead us to believe that the supplementary materials were not provided by the journal during the review process. We have contacted the journal to ensure that supplementary materials will be accessible for the reviewers in this second round of review.
POINT 10: Line 227: Information on missing data (e.g., percentage of missing for each variable) should be made transparent for readers.
RESPONSE 10: We now provide a detailed report on the data for each variable and the relevant analyses to ensure the quality of the imputations in the supplementary material. We hope that in the present round of review, this material will be accessible to reviewers so that they will be able to verify all the information relating to the imputation process.
POINT 11: Line 241: What were the qualitative variables assessed? Qualitative data indicate responses involving text.
RESPONSE 11: We have corrected this error.
POINT 12: Discussion - Line 441: It will be great if the authors can insert a paragraph here to discuss the implication of their findings for policy makers in Spain.
RESPONSE 12: We have modified the text to discuss the implication of our findings for policy makers in Spain.
“Multilevel system-strengthening approaches that integrate mental healthcare into HIV care and prevention within healthcare and community-based organizations need to be addressed in Spain and to be tailored to specific subgroups of PLWH that are at a higher risk of depressive symptoms.”

Round 2
Reviewer 2 Report
I hope the authors have enjoyed the reviewing process and found it beneficial to improve the paper.
I have two final comments for the authors to address:
1. Given that the research team used a single-item question to measure gender, its limitation of falling short of detecting trans people who simply identify as 'men' or 'women' should be noted. See Gloria Fraser's paper: https://doi.org/10.1080/19419899.2018.1497693
2. The phrase 'gender perspective' in Lines 111 and 115 needs to be clarified. I understand this as examining gender differences for determinants of depressive symptoms?
Author Response
We are very grateful for the reviewer's comments. They have helped us to raise gender-relevant issues that we will apply in future studies.
We have therefore addressed the two comments:
Regarding comment 1, we have added this aspect to the limitations of the study together with the bibliographical reference to Gloria Fraser's paper.
"Fifth, we used a single-item question to measure gender. Although different categories were available (woman, man, trans woman, trans man, genderqueer, and other), this might be insufficient for the detection of trans people. Future studies should include categorical lists where the participant can select more than one option, as well as include an open-text option where the person can report their gender identity [64]."
And concerning comment 2, we have modified the text to clarify that what we were examining gender differences for determinants of depressive symptoms.
"Therefore, there is a need to investigate gender group differences by considering the multiple factors associated with mental health and gender so that the problems affecting this population can be addressed in a timely manner [26].
The objective of the present study was to identify risk and protective factors for sociodemographic variables, comorbidities, health-related behaviours, and social environment–related variables associated with depressive symptoms in PLWH, taking into account gender differences, in order to enable clinical and public health interventions."
We hope that these changes have improved the quality of our work. We would be happy to make any further changes in this respect if the reviewer feels that they are necessary.